# Development and Comparison of Various Coated Hard Capsules Suitable for Enteric Administration to Small Patient Cohorts

**DOI:** 10.3390/pharmaceutics14081577

**Published:** 2022-07-29

**Authors:** Nicole Fülöpová, Sylvie Pavloková, Ivan DeBono, David Vetchý, Aleš Franc

**Affiliations:** 1Department of Pharmaceutical Technology, Faculty of Pharmacy, Masaryk University, 612 42 Brno, Czech Republic; 507073@muni.cz (N.F.); pavlokovas@pharm.muni.cz (S.P.); vetchyd@pharm.muni.cz (D.V.); 2Department of Pharmacy, Mater Dei Hospital, 2090 Msida, Malta; ivan.debono.94@um.edu.mt

**Keywords:** DRcaps^TM^ capsules, hard gelatin capsules, Eudragit^®^ S, enteric coating, immersion method, principal component analysis

## Abstract

Pharmaceutical technology offers several options for protecting substances from acidic environments, such as encapsulation in enteric capsules or dosage form with enteric coating. However, commercial enteric capsules do not always meet limits for pharmacopeial delayed release, and the coating process is generally challenging. Preparing small enteric batches suitable for clinical use is, therefore, an unsolved problem. This experiment offers a simple coating process of DRcaps^TM^ capsules based on hypromellose (HPMC) and gellan gum to achieve small intestine administration. In addition, DRcaps^TM^ capsules were compared to hard gelatin capsules to evaluate the suitability of the coating method. Both capsules were immersed in dispersions of Eudragit^®^ S 100, Acryl-EZE^®,^ and Cellacefate at concentrations of 10.0, 15.0, and 20.0% and dried. Coated capsules were evaluated by electron microscopy, disintegration, and dissolution test with a two-step pH change (from 1.2 to 6.8, then to 7.5) to simulate passage through the digestive tract. DRcaps^TM^ capsules coated with Eudragit^®^ S and Cellacefate achieved acid resistance. While samples coated with Eudragit^®^ S released their contents within 360 min at pH 6.8 (small intestine), regardless of polymer concentration, capsules with 15.0 and 20.0% coatings of Cellacefate released content at pH 7.5 (colon) within 435 and 495 min, respectively.

## 1. Introduction

Capsules are commonly used in the pharmaceutical industry due to their uncomplicated manufacture as a simple dosage form for oral delivery [1]. Conventional hard capsules are usually made of gelatin, a polymer degrading within a few minutes after swallowing [2]. However, many drugs are labile in the acidic area of the stomach. Therefore, it is a focus of much research to target digestive tract environments behind the stomach, where they could be absorbed over a larger area in a more pH-acceptable environment [3]. Enteric capsules could be a suitable dosage form for this purpose. They should be able to resist an acidic pH of 1.2 (gastric conditions) but should disintegrate at a pH of around 6.8 (intestinal environment) or potentially around pH 7.5 (colon environment) [4,5].

Recently, there has been an effort to produce enteric capsules for small patient cohorts, especially in clinical trials for administering drugs that degrade in an acidic environment or for various drugs for colon targeting dedicated to treating Crohn’s disease and colorectal cancer, such as mesalazine, prednisone, and budesonide, or for oral vaccination. It is also suitable for administering fecal bacteria through a process commonly known as fecal microbiota transplantation (FMT), which should be targeted to the distal parts of the digestive tract, preferably the colon [6,7,8,9,10]. The research focused on making their preparation process reproducible, adapted to the individual patient, and feasible even in laboratory conditions [11]. A possibility in how to solve this problem is by using commercially produced enteric (including colonic) capsules. Therefore, enteric capsule drug delivery technology (ECDDT) was developed to provide oral delivery with enteric protection by incorporating pharmaceutically approved enteric polymers into the capsule shell without using coatings [12]. Capsugel^®^ (Lonza Company, Colmar, France) launched DRcaps^TM^ capsules consisting of 95% HPMC and 5% gellan gum [13]. According to the company, DRcaps^TM^ disintegrates 45 min later than conventional hard gelatin capsules in the dissolution test and entirely in the small intestine [14]. However, several studies’ results did not confirm their acid resistance [15,16]. CapsCanada (Windsor, ON, Canada) introduced enteric AR-CAPS^®^ capsules [16] made of HPMC and hydroxypropyl methylcellulose phthalate (HPMCP). According to the producer, capsules released their content in the intestinal part without further coating [17]. Nevertheless, a related study shows that capsules did not meet the pharmacopeial limits for delayed-release dosage form [16,18]. Suhueng Capsule (Seoul, South Korea) produced acid-resistant EMBO CAPS^®^ AP capsules of HPMC and pectin [19,20]. Still, EMBO CAPS^®^ AP capsules released more than 10% of capsule content during the first 120 min of the dissolution test in the acidic stage [5,20], which does not meet the requirements of the Ph. Eur. 2.9.3 [18]. Company BioCaps^®^ (El Monte, CA, USA) manufactured fully acid-resistant HPMC capsules BioVXR^®^ [5,21]. The associated patent also mentions using polymers, such as pectin, propylene glycol alginate, or xanthan gum and gelling agents, such as gellan gum or carrageenan [22]. However, the related study has shown that the drug release profile from the BioVXR^®^ capsules did not comply with the pharmacopeia requirements for delayed release [5,18]. Therefore, ECDDT is not always a reliable technology for achieving acid resistance and cannot be reliably used for colon targeting.

Another way to protect the capsule against the acidic environment is enteric coating. In this process, particular polymers are applied to the surface of the capsule by various methods [23]. In addition, this approach could create a suitable dosage form for transport to more distant parts of the digestive tract [24]. The enteric polymers commonly used are poly-(methacrylic acid-co-methyl methacrylate) under the trade name Eudragit^®^ S, L, and FS from Evonik Industries (Essen, Germany) [25]; methacrylate mixtures as Acryl-EZE^®^ manufactured by Colorcon^®^ (Colorcon^®^ Limited, Dartford, UK) [26]; cellulose-based polymers as hydroxypropyl methylcellulose acetate succinate (HPMCAS); HPMCP, Cellacefate or polyvinyl derivates, such as PVAP (polyvinyl acetate phthalate); and others. [23,24,27]. However, coating hard gelatin capsules is a time-consuming process involving special types of equipment (coating pan or fluid bed coater), which is not available in hospital facilities where, for example, capsules containing FMT are prepared [28]. In addition, the process has several complications, such as a smooth surface of capsules, where polymers do not adhere properly [24], softening of the capsule wall after water-based polymers are used [28], or embrittlement of the capsule during the drying process [7]. Various studies addressed these problems using immersion in polymer dispersion as a method for enteric coating with promising results without using any other equipment. Even colon-targeted administration was achieved with this simple method. However, repeating this immersion procedure 2–3 times and adding a plasticizer to the polymeric dispersion is required [29,30]. The plasticizer is usually needed to improve the film’s mechanical properties. When the film composition contains more than one component, the compatibility of all components is important to ensure the stability of the capsule shell [31]. Plasticizers in the hard capsule shell could sometimes lead to phase separation of the already applied coating [32]. The polymer coating quality could also be affected by how plasticizers are incorporated into the polymer dispersion [33]. The process of coating hard gelatin capsules entails several complications, so a strategy to coat an already modified capsule, e.g., the DRcaps^TM^ capsule, with a suitable polymer dispersion where this combination could have ideal synergistic properties seem a viable option.

This experimental work aimed to develop enteric capsules designed for the simple and reproducible preparation of small batches in the laboratory, intended for clinical/preclinical studies requiring acid-resistant delivery technology. The immersion method was chosen as an inexpensive and time-saving method of coating. Eudragit^®^ S (dissolves at pH > 7); Acryl-EZE^®^ (soluble at pH > 5.5) and Cellacefate (soluble at pH ≥ 6.2) were selected as promising enteric polymers [23,25,26,34]. Polymer dispersions in three concentrations (10.0, 15.0, and 20.0%) were applied to DRcaps^TM^ capsules once. For reproducibility and facilitated production feasibility of small batch preparation of enteric capsules in laboratory conditions, the method of preparation without adding a plasticizer was chosen. The coated hard gelatin capsules were also prepared with the same method as DRcaps^TM^ to compare the suitability of the coating process and chosen enteric polymers. Capsules were subjected to a dissolution test with continuous pH change and a disintegration test. The film properties were examined using scanning electron microscopy (SEM). Statistical interrelationships between the formulation variables and measured capsules characteristics were investigated using the analysis of variance (ANOVA), principal component analysis (PCA), and metrics for comparing dissolution curves.

## 2. Materials and Methods

### 2.1. Materials

The caffeine (Dr. Kulich Pharma, Hradec Králové, Czech Republic) was used in the dissolution test as a release indicator. Caffeine is released independently of pH; the release profile will not be disturbed by its decomposition in the dissolution medium. Therefore, caffeine is standardly used as a model drug in these cases. Lactose monohydrate (Dr. Kulich Pharma, Hradec Králové, Czech Republic) was used as capsule filler. Eudragit^®^ S (Evonik Industries AG, Essen, Germany), Cellacefate (Sigma-Aldrich, St. Louis, MO, USA), and Acryl-EZE^®^ (Colorcon^®^ Limited, Dartford, UK) were used as pH-responsive polymers for acid-resistant capsule coating. These products are approved for pharmaceutical use. Acetone (Sigma-Aldrich, St. Louis, MO, USA), Ethanol 96% (*w*/*w*) (Sigma-Aldrich, St. Louis, MO, USA), and purified water (prepared in-house) were used as solvents for the suitable polymer. DRcaps^TM^ capsules (Capsugel^®^, Bornem, Belgium) and hard gelatin capsules (Dr. Kulich Pharma, Hradec Králové, Czech Republic) were used as the pharmaceutical dosage form. The chemicals used for the preparation of dissolution media were as follows: for disintegration test—sodium chloride; 0.1M hydrochloric acid (HCl) for preparation of 1.2 pH gastric juice without pepsin (2 g/80 mL per 1000 g); potassium dihydrogen phosphate; sodium dihydrogen phosphate dodecahydrate (138.72 g/350.84 g per 1000 g) for buffer with pH 6.8; for the dissolution test, the same pH 1.2 gastric juice as for the disintegration test was used; sodium triphosphate for adjustment pH of 1.2 pH buffer to pH 6.8 and after that to pH 7.5 (all Sigma-Aldrich, St. Louis, MO, USA). 

### 2.2. Preparation of Enteric-Coated Capsules

The powder mixture, consisting of lactose monohydrate and caffeine (400 mg and 100 mg per capsule), was blended by a homogenizer Turbula T2C (WAB, Basel, Switzerland) at 40 rpm for 30 min. The caffeine mixture was then dispensed into DRcaps^TM^ and hard gelatin capsules (size 0) with a capsule filling machine (laboratory device, HEROS^®^, Olomouc, Czech Republic). Each sample consists of 20 capsules. Weight uniformity and drug content of capsules were evaluated according to Ph. Eur. 2.9.5 (Uniformity of mass of single-dose preparations.) and Ph. Eur. 2.9.6 (Uniformity of content of single-dose preparations) [18].

For capsule coating, polymer dispersions were prepared in three concentrations (10.0%; 15.0%; 20.0% (*w*/*w*)) consisting of an enteric polymer and a suitable solvent according to available resources; Cellacefate in acetone [18,34]; Eudragit^®^ S in Ethanol 96% (*w*/*w*) [25] and Acryl-EZE^®^ in purified water [26]; using stirrer Heidolph RZR 2020 (Heidolph Instruments GmbH & Co. KG, Schwabach, Germany) at 200 rpm for 60 min. Due to the reproducibility and easy feasibility of individual preparation of enteric capsules in laboratory conditions, all variables were reduced to a minimum. For these reasons, the preparation method without adding a plasticizer has been chosen. Filled capsules were immersed in polymer dispersion only once, followed by drying at room temperature 24 °C for 24 h. Subsequently, the capsules were evaluated by predetermined tests. The characteristic of prepared hard capsule samples is presented in Table 1.

### 2.3. Viscosity Measurement

Viscosity measurement of polymer dispersions was performed on a rotary viscometer Brookfield DV-II+ Pro (Brookfield, London, UK) using a tempered constant temperature attachment (26.0 ± 0.5 °C) and SC4 21 spindle suitable for less viscous samples (25–500,000 cP). Measurement results were recorded using Rheocalc software (Brookfield, London, UK) [35]. After switching on the device, the thermometer was connected. Subsequently, the adapter was filled with a 7.1 mL sample, and the SC4 21 spindle was inserted. Then, using Rheocalc software, a higher spindle rotation with 200 rpm was started. After stabilization, the viscosity value was read. The measurement was repeated three times for every polymer dispersion. Finally, the mean values of viscosity (cP) with standard deviations (SD) and torque (%) ± SD were calculated.

### 2.4. Weight Gain of the Coating

All capsule batches were weighed twice; before and after the coating process on an analytical scale KERN 870–13 (Gottl. KERN & Sohn GmbH, Balingen, Germany). The average weights of the polymer coating (mg) ± SD were calculated based on the difference in weight before and after the coating process. In addition, the mean weight gain of the coating layer (%) ± SD was also determined for each sample.

### 2.5. Capsule Structure Images

A scanning electron microscope (SEM; MIRA3, Tescan Orsay Holding, Brno, Czech Republic) equipped with a secondary electron detector was primarily used to observe the coating of capsules. The sample preparation for the microscopic measurement included a scalpel cross-section of the capsule and mounting the specimen on a SEM stub using a conductive carbon double-faced adhesive tape (Agar Scientific, Essex, UK). The next step was coating with a 20 nm gold layer using the metal sputtering coating method in the argon atmosphere (Q150R ES Rotary-Pumped Sputter Coater/Carbon Coater, Quorum Technologies, Laughton, UK). The SEM images were obtained at an accelerating voltage of 3 kV and various image view fields, while the image view field of 500 μm (or 1000 μm in one case) was chosen to observe the thickness of the capsule coating. The measurement was repeated 20 times from multiple images of each sample. DRcaps^TM^ and hard gelatin capsules without coating were also examined using SEM to determine the relations between the surface of uncoated capsules and polymer coating adhesion in the case of coated samples.

### 2.6. Disintegration Test

According to the Ph, Eur. 2.9.1 (disintegration of tablets and capsules), enteric-coated capsules were evaluated in an Erweka ZT4 instrument (Erweka GmbH, Langen, Germany) [18]. A disintegration test was performed first in a pH 1.2 (0.1M HCl) simulated gastric environment for 120 min, followed by pH 6.8 (phosphate buffer), simulated intestinal environment. The volume of both media was 900 mL, preheated at 37.0 ± 0.5 °C. Disintegration tests were performed with disks unless otherwise specified. The device worked until the capsules completely disintegrated. Requirements for a delayed-release dosage according to Ph. Eur. 2.9.1 were applied, which was the minimum acceptable benchmark. None of the six capsules should show signs of disintegration or cracks, allowing the contents to leak for 120 min in an acidic medium. At pH 6.8, capsules should disintegrate within 1 h to meet the requirements of the delayed-release dosage form. If disintegrated later, they are likely suitable for targeting distal parts of the intestine. Capsule disintegration time ± SD (min) was monitored.

### 2.7. Dissolution Test

The dissolution test was performed on the USP apparatus 2 (Sotax AT-7, Donau Lab, Zurich, Switzerland) using the paddle method at 50 rpm. Six capsules from every batch (see Table 1) were placed in sinkers to avoid flotation on the media’s surface. Capsules were first established in 900 mL of pH 1.2 acidic medium (0.1M HCl) preheated at 37.0 ± 0.5 °C for 120 min to simulate conditions in the digestive tract. After 120 min, the pH was increased to 6.8 by adding 18.7 g of sodium triphosphate per vessel. The pH was modified to 7.5 by adding 5.8 g of sodium triphosphate per vessel to simulate the colonic area 360 min later. The sampling points were set at 15 min intervals until the total duration of the 12h dissolution test. The dissolved content of caffeine from capsules was analyzed by automatic UV/Vis analysis at wavelength 275 nm. The dissolution test is characterized by the dissolution profile of caffeine release for each sample, where sampling points are plotted after 60 min. The dissolution profile is shown as the mean value calculated from six units for each sample ± SD (min) displayed as error bars.

### 2.8. Comparison of Dissolution Profiles

The similarity factor *f*_2_ and dissolution efficiency (DE) was used to compare the dissolution profiles of capsule batches. The similarity factor *f*_2_ was calculated using the well-known formula [36]. Similarity factor *f*_2_ is not given in any unit; it only indicates the degree of conformity between the individual dissolution profiles. The profiles are similar if *f*_2_ values are between 50 and 100. Dissolution efficiency (DE) is the area under the dissolution curve between two-time points expressed as a percentage of the curve at maximum dissolution over the same period [37]. The trapezoidal method was used to calculate the area under the curve (AUC) for each sample’s average dissolution profile. The dissolution profiles were compared using the metrics ΔDE, which is the percentage ratio of the DE value of one dissolution profile to the DE value of another. The reference and the test product are equivalent if the ΔDE is within appropriate limits (±10%) [38]. Both metrics were calculated to observe the differences in active pharmaceutical ingredients (API) release during the dissolution test. The effect of each parameter (capsule type, polymer type, polymer concentration) was separately assessed when setting the other two parameters at a constant level. The calculation was performed for all sample pairs within the influence of the considered parameter.

### 2.9. Statistical Analysis

The statistical significance of the effects of formulation parameters (concentration and type of polymer as well as the type of capsule) on measured properties of dispersion/capsule were investigated through ANOVA. The viscosity of coating dispersion and final capsule features selected dissolution characteristic (released amount of API at 120 min) and coating characteristics (weight gain and thickness determined by SEM) were considered output variables. According to pharmacopeial limits for delayed-release formulations applied during the dissolution test, none of the individual values should exceed 10% of API released at 120 min in pH 1.2 to be considered acid resistant. Therefore, a dissolution time of 120 min was selected as a crucial parameter of the dissolution test. The ANOVA model assessed all main effects and higher-order interactions. The results are presented in *p*-values, describing the significance of the relevant factors, statistically significant effect in the case of *p* < 0.05, or insignificant effect for *p* > 0.05. The ANOVA outputs were also examined by visualization using interaction plots. A PCA on standardized data was employed to assess the combined effect of formulation parameters on the various responses. A mutual comparison of the subsequent PCA visualization via loadings plot and scores plot was investigated to describe the interrelationships of variables in the data set. Data analysis was performed using R software, version 4.0.1 [39].

## 3. Results and Discussion

In this experimental study, 18 batches of enteric-coated capsules were prepared and evaluated to investigate whether the immersion of capsules in polymer dispersion is an acceptable method for achieving acid resistance. The suitability of this method was compared between different polymer dispersions (Acryl-EZE^®^, Cellacefate, and Eudragit^®^ S) in various concentrations (all in 10.0%, 15.0%, and 20.0%) applied at both types of capsules (DRcaps^TM^ and hard gelatin). This preparation method could be suitable for samples in small quantities without economically demanding equipment. Furthermore, the method without adding a plasticizer was chosen to simplify and reproduce the process under laboratory conditions and reduce possible variable parameters. Coated DRcaps^TM^ capsules and hard gelatin capsules were also compared to determine relations between capsule shells and different polymer dispersions in various concentrations. 

According to Ph. Eur., all batches of capsules met the uniformity requirements of single-dose preparations with a drug content of 95.60–105.90% of the theoretical amount of 100 mg caffeine (see Table 1) [18]. In addition, the weight uniformity of capsules results ranged from 423.63 ± 9.05–529.19 ± 8.05 mg by hard gelatin capsules and 432.36 ± 6.45–528.25 ± 7.97 mg by DRcaps^TM^ capsules (see Table 1).

### 3.1. Viscosity Measurement

In general, viscosity could be defined as a fluid’s resistance to deforming under the influence of shear stresses. It is manifested by internal friction and depends on various parameters, including temperature [40]. Due to this fact, the viscosity of polymer dispersion was measured at the same temperature (26.0 ± 0.5 °C) to eliminate the temperature-related cofounding factor. According to operating manual instructions for rotary viscometer Brookfield DV-II+ Pro issued by Brookfield (London, UK), the torque values from 10 to 100% are recommended. The same test methodology (spindle size, rotation speed, time, and container) also impacted obtaining comparable results between all polymers dispersions, so all prepared dispersions were evaluated under the same conditions [35]. Same measurement conditions lead to low viscosity values for Acryl-EZE^®^ dispersions with no significant differences in increasing polymer concentration 1.3 ± 0.12–3.5 ± 0.00 cP (see Table 2). Dispersions formed by Eudragit^®^ S and Cellacefate polymers have higher viscosity values with more significant differences in obtained results as Acryl-EZE^®^ dispersions. With increasing dispersion concentration, viscosity values significantly increase for Cellacefate from 28.8 ± 2.16 to 176.2 ± 21.23 cP and for Eudragit^®^ S from 36.1 ± 1.25 to 216.3 ± 6.60 cP (see Table 2). The increasing concentration of film polymer in organic dispersion leads to a more significant increase in viscosity than in the case of the aqueous dispersion. The results correspond to literary data [41].

### 3.2. Characteristics of Polymer Coating: Weight Gain of Coating; Thickness of Coating; Capsule Structure Images

Several parameters characterize the resulting polymer capsule coating: film thickness (μm) and weight of the coating (mg), converted to a percentage gain of the coating (see Table 3). For all samples, selected SEM images displayed the coating thickness (see Figure 1, Figure 2 and Figure 3). SEM images were obtained under the same conditions (3 kV; 500 μm). Only sample Ge_Ac20 was scanned at different conditions (3 kV; 1000 μm) because it was impossible to measure film thickness correctly on the higher magnification images (see Figure 2). The separation of the coating layer observed on SEM images (see Figure 1, Figure 2 and Figure 3) may have occurred due to the pressure exerted on the capsule during the incision. However, this does not seem to affect the final coating thickness measurement. The thickness of the individual coatings relates to their weight (with increasing thickness, the calculated weight of coating also increases) is shown in Table 3 and confirmed by nonparametric Spearman’s correlation (R_s_ = 0.69, *p* = 0.002). For coated DRcaps^TM^, capsules were determined by weight gains of coatings as follows (see Table 3), Acryl-EZE^®^ coating values were between 9.49 ± 2.44 and 17.18 ± 1.80 (mg); for Cellacefate in the range of 6.28 ± 0.86–18.49 ± 1.91 (mg) and Eudragit^®^ S values were from 7.99 ± 0.93 to 23.78 ± 4.47 (mg). In addition, thickness values were determined from SEM images; for Acryl-EZE^®^ coatings from 19.40 ± 3.79 to 35.60 ± 7.97 (μm); for Cellacefate coatings in the range of 28.35 ± 3.12–70.00 ± 2.71 (μm) and Eudragit^®^ S coatings were from 5.70 ± 0.73 to 37.75 ± 2.67 (μm), respectively. For hard gelatin capsules coated by enteric dispersions, the values of weight gain were as follows (see Table 3): Acryl-EZE^®^ from 5.47 ± 1.76 to 15.09 ± 1.82 (mg); for Cellacefate in the range of 3.39 ± 0.68–18.44 ± 1.74 (mg); and Eudragit^®^ S were from 4.28 ± 0.48 to 9.85 ± 1.29 (mg). Therefore, the values of coating thickness were as follows: Acryl-EZE^®^ from 12.50 ± 3.12 to 57.90 ± 6.83 (μm); Cellacefate in the range of 8.35 ± 1.14–23.20 ± 3.74 (μm); Eudragit^®^ S in range of 11.10 ± 2.20–19.80 ± 2.55 (μm). The different effect of polymer concentration on the increase in coating quantity in the case of Acryl-EZE^®^ compared to Eudragit^®^ S may be due to the presence of pigments and talc in the dispersion of Acryl-EZE^®^. Their presence may lead to a decrease in the adhesion of the polymer to the capsule shell, which could limit the increase in coating thickness [42,43]. According to these results, the weight of the resulting coating also increases with the increase in polymeric concentration dispersion (see Table 3).

According to previous studies, the adhesion of the polymer coating is likely to depend on the capsule’s surface structure, where the gelatin capsule’s surface is smooth, and that of the HPMC capsule is rough [7,24]. However, from the obtained SEM images (see Figure 4A,B), the surface of the gelatin capsules is rougher than the HPMC/Gellan capsules. Nevertheless, the adhesion of the polymer coating remains the same as in the previous studies—better to the HPMC/Gellan capsules than to the hard gelatin capsules (see Figure 1, Figure 2 and Figure 3) regardless of the type of applied polymeric dispersion. Most visible differences are between samples DR_Ac20/Ge_Ac20; DR_Ce15/Ge_Ce15; DR_Eu10/Ge_Eu10. Results obtained in Table 3 also support these assumptions—lower weight and thickness of coating applied on hard gelatin capsule than on DRcaps^TM^ capsule. These results indicate that the adhesion of the polymer coating to the capsule shell is not affected by its surface’s structure but by the capsule’s composition.

### 3.3. Disintegration Test

During the disintegration test, the rupture of both capsules (DRcaps^TM^ and hard gelatin capsules) was visually evaluated, keeping pharmacopeial limits as reference [18]. For a simple evaluation of the disintegration test, the total disintegration time of capsules was considered as the time when the capsules showed the first signs of rupture [16].

As seen in the plot (Figure 5B), the immersion method is mostly not sufficient for almost all batches of hard gelatin capsules (Ge_Ac10; Ge_Ac15; Ge_Ac20; Ge_Eu10; Ge_Eu15; Ge_Eu20; Ge_Ce10 and Ge_Ce15), since the capsules disintegrated within 120 min in an acidic environment (10 ± 1 min; 14 ± 1 min; 32 ± 2 min; 13 ± 3 min; 13 ± 4 min; 30 ± 4 min; 49 ± 16 min and 87 ± 11 min, respectively). Only sample Ge_Ce20 obtained values within pharmacopeial limits with a 129 ± 7 min disintegration time. Results of the disintegration test of DRcaps^TM^ capsules show (see Figure 5A) that the two batches (DR_Ac10 and DR_Ac15) did not meet the pharmacopeial limits with a disintegration time of 73 ± 11 min and 73 ± 10 min. The other two samples (DR_Ac20 and DR_Ce10) disintegrated within limits (148 ± 12 min and 159 ± 6 min). On the other hand, samples (DR_Ce15; DR_Eu10; DR_Ce20; DR_Eu15 and DR_Eu20) did not disintegrate in the specified range of 120–180 min indicated for the delayed-release dosage form. Their disintegration occurred later (188 ± 9 min; 200 ± 19 min; 210 ± 4 min; 258 ± 21 min; 314 ± 29 min, respectively). This could indicate the possibility of reaching a more distant part of the digestive tract according to the transit time of the dosage form in the digestive tract [44].

### 3.4. Dissolution Test. Similarity Factor f_2_ and Dissolution Efficiency (Expressed as ΔDE) of Compared Dissolution Profiles

According to the Ph. Eur. 2.9.3., a batch of capsules met the requirements for the delayed-release dosage form if none of the six measured samples in an acidic environment (pH 1.2; 0.1M HCl) exceeded 10% release of the API. After changing pH to 6.8 for simulating the intestinal environment, the dosage form should release content within one hour. The dissolution end time (total dissolution time) is given by the general lower limit of “Q” (amount of released API 75 ± 5%) [18]. Prepared samples were evaluated according to these requirements. Complete dissolution time was also monitored. Based on it, the full release of the dosage form in a particular segment of the digestive tract can be expected. During the passage of the dosage form through the digestive tract, dynamic changes occur. Transit time and pH values are approximately in the stomach for 0.25–2 h and pH 1.0–2.5; in the small intestine for 3–4 h and pH 6.0–7.8; in the colon for 6–48 h and pH 5.5–8.0. [6,8,44].

None of the coated hard gelatin capsule batches met the pharmacopeia limits because the released amounts of the API were more than 10% in 120 min at acidic environment (pH 1.2) See Figure 6D: Ge_Ac 10 (93.74 ± 3.24%); Ge_Ce10 (61.82 ± 16.14%) and Ge_Eu10 (82.68 ± 12.32%); Figure 6E: Ge_Ac15 (94.43 ± 5.02%); Ge_Ce15 (33.90 ± 17.89%); Ge_Eu15 (91.36 ± 6.69%); Figure 6F: Ge_Ac20 (86.05 ± 8.35%); Ge_Ce20 (14.23 ± 15.19%); Ge_Eu20 (43.10 ± 18.76%), respectively. Coated DRcaps^TM^ capsules met the pharmacopeia requirements, except for samples DR_Ac10; DR_Ac15 (see Figure 6A,B). The enteric coating formed by a low polymer concentration was also sufficient without plasticizer, except for Acryl-EZE^®^ dispersions. Colorcon^®^ (Colorcon^®^ Limited, Dartford, UK) declares the acid resistance of Acryl-EZE^®^ coatings if a plasticizer is incorporated into the coating system [45]. However, sample DR_Ac20 is acid resistant even without the presence of a plasticizer (the amount of released API in 120 min was 2.37 ± 0.32%). In addition, a correlation between results obtained from the disintegration test and dissolution test was with DR_Ac samples, also confirmed as in previous studies [46]. Samples DR_Ac10 and DR_Ac15 did not pass both tests, and DR_Ac20 met the pharmacopeial requirements for both tests). All DR_Ce samples did not fully comply with the pharmacopeia limits [18]. The amount of API detected at 120 min was: 2.98 ± 0.90 %, 1.67 ± 0.38 %, and 0.42 ± 0.44% (see Figure 6A–C).

As the concentration of polymer dispersion used for coating increases, the total dissolution time within the polymer dispersion type (Cellacefate) also increases, namely: DR_Ce10 released 82.06 ± 5.58% API in 270 min; DR_Ce15 in 435 min 75.33 ± 15.54% API; DR_Ce20 in 495 min 78.86 ± 20.92% API. The dissolution profiles of the Cellacefate-based samples are affected by the better continuous increase of the polymer coating, which is related to its increasing concentration in the coating dispersion. A possible explanation is the better cohesion and compactness of Cellacefate coatings compared to other polymeric materials based on cellulosic, methacrylic, or vinyl derivates [47].

DRcaps^TM^ capsules coated with Eudragit^®^ S provided the best results during the dissolution test for achieving the small intestine (see Figure 6A–C). Within 120 min, the samples DR_Eu10 released 1.69 ± 0.40%; DR_Eu15 released 0.45 ± 0.35%, and the DR_Eu20 was the only sample that released no API (0.0 ± 0.0%). After the pH change to 6.8, the DR_Eu20 sample released API as first from DR_Eu samples. Complete dissolution time with the smallest SD (82.20 ± 4.27% in 315 min) indicates that all six capsules of this batch dissolved simultaneously in an environment simulating the small intestine. The DR_Eu10 and DR_Eu15 samples released their content (75.45 ± 25.27%; 77.33 ± 27.88% in 360 min) with a higher SD. According to obtained results from the dissolution test (see Figure 6), DRcaps^TM^ capsules coated by 20.0% polymeric dispersion of Eudragit^®^ S by an immersion method without the addition of plasticizer (no release of API in an acidic medium within 120 min) seem to be the suitable enteric-coated dosage form. However, DRcaps^TM^ capsules coated with polymer dispersion of Cellacefate at concentrations of 15.0% and 20.0% have dissolution profiles that are suitable even for drug delivery to more distant parts of the digestive tract, with total dissolution times at 435 min and 495 min. Therefore, this technology could be suitable for treating Crohn’s disease or colorectal carcinoma or administering FMT to the colon area [6,8,9,44].

The difference between dissolution profiles was also observed through similarity factor *f*_2_ and calculation of ΔDE in Table 4. Values presented in Table 4 (A) and (B) reveal that the profiles are not similar in all cases except for four batches of combination (DR_Ce15/DR_Ce20; DR_Eu10/DR_Eu15; Ge_Eu10/Ge_Eu15 and DR_Eu20/DR_Ac20), where the *f*_2_ exceeded the value of 50 (79.44, 66.12, 53.72, and 52.19, respectively). These results suggest that the dissolution profiles of DRcaps^TM^ capsules, or dosage forms with Eudragit^®^ S polymer in lower concentrations, regardless of which capsule the polymer is applied to, are similar. However, both values (*f*_2_; ΔDE) indicate that the release from dosage forms formed by the same polymeric coatings applied to different capsules (hard gelatin and DRcaps^TM^ capsules) is not similar, as can be seen in Table 4 (C). The similarity factor *f*_2_ is in the range of 6.14 (DR_Eu10/Ge_Eu10) to 15.28 (DR_Ce10/Ge_Ce10) and ΔDE, even in the range from 49.85 (DR_Ac10/Ge_Ac10) to 1546.18% (DR_Ce15/Ge_Ce15). Therefore, the capsule type plays a significant role in the release rate.

### 3.5. Statistical Analysis

The outputs of ANOVA testing are summarized in Table 5. As can be seen from the *p*-values, the significant effects of all tested factors (polymer type, concentration, and capsule type) and second- and third-order interactions were revealed. These findings are also visible from the interaction graphs (see Figure 7), where increasing/decreasing trends of the curves depending on the polymer concentration occur. In addition, the differences in the given quantity values (viscosity, coating gain/thickness, and released amount of API) for different polymer and capsule types can also be observed. Moreover, the mutual interactions can be inferred from the different shapes of the curves in the graphs or the presence of several crossing curves. ANOVA testing of the influence of these variables in individual subgroups differentiated by polymer type, polymer concentration, or capsule type confirmed similar trends as general testing involving all samples. Associated *p*-values of all parameters were lower than 0.05, and almost all mutual interactions between factors were also significant (specific *p*-values are not included in the article).

The viscosity of polymer dispersion increases in the order: Acryl-EZE^®^ < Cellacefate < Eudragit^®^ S, especially at two higher polymer concentrations (see Figure 7A). Furthermore, a significant increase in viscosity with the polymer concentration was corroborated. A meager but statistically significant viscosity increase was found for Acryl-EZE^®^, while for Cellacefate and Eudragit^®^ S, a well recognizable rise was observed. Primarily, a higher polymer concentration (15.0 or 20.0%) in the Cellacefate and Eudragit^®^ S samples increases the viscosity. An increase in the released amount of API for hard gelatin capsules was detected in the order: Cellacefate < Eudragit^®^ S < Acryl-EZE^®^ (see Figure 8B). Samples DR_Ce and DR_Eu capsules differ from all other samples. At 120 min of the dissolution time, these batches have a significant lag time in all polymer concentration levels. Furthermore, a smaller amount of API at 120 min for higher polymer concentration was observed, especially for samples DR_Ac and Ge_Ce. Therefore, the higher amount of API in hard gelatin capsules than in DRcaps^TM^ capsules is unquestionable. In capsule coating, the polymer concentration was evaluated as the dominant effect. The direct relationship between the polymer concentration and coating gain and thickness is visible (see Figure 7C,D). However, the increase in coating thickness is less noticeable than in coating gain. When comparing the coating gain/thickness depending on the capsule type, observable differences between the two groups (DRcaps^TM^ > hard gelatin capsules) were identified. A high difference in coating gain was found, especially for samples Eudragit^®^ S 15.0% and 20.0%. Polymer type also proved to be a statistically significant parameter, but there is also a considerable effect of interactions; the dependencies are more complex because they differ according to the capsule type and polymer concentration. The highest values of coating gain were reached for the samples DR_Eu15, DR_Eu20, DR_Ac15, and DR_Ac20. The coating of the most increased thickness was detected in the samples DR_Ce20; Ge_Ac15, and Ge_Ac20.

The PCA verified the conclusions mentioned above and provided insight into the main trends in the data structure. The resulting PCA model’s first two principal components described 83.7% of total variability together. Graphical representations of PCA have been created: a loadings plot (see Figure 8A) and a scores plot (see Figure 8B). There is a considerable variation in the monitored properties of coating dispersions and capsules with different formulation parameters settings (see Figure 8B). Polymer type’s contribution to the data set’s variability is evident. In the scores plot (see Figure 8B), the clustering of the samples based on the polymer type is observable in this arrangement: Acryl-EZE^®^ on the bottom, Cellacefate slightly above the center of the coordinate system, and Eudragit^®^ S at the top. In the corresponding bottom-up direction in the loadings plot (see Figure 8A), the effect of the polymer dispersion viscosity is dominant. A considerable difference between the Acryl-EZE^®^ samples and capsules with other polymer types can be observed, as the corresponding points in the scores plot are far from each other. The low viscosity of the Acryl-EZE^®^ dispersion caused this difference. The polymer concentration increases in the direction from right to left (see Figure 8B), which corresponds to the increase in the weight gain of the coating, coating thickness, and a decrease in the released amount of API at 120 min (see Figure 8A). Depending on the capsule type, the grouping of objects showed that the DRcaps^TM^ specimens are more towards the upper left quadrant. In contrast, hard gelatin capsules occupy the space in the lower right direction. In the loadings plot (see Figure 8A), the influence on the released amount of API corresponds to this direction, so DRcaps^TM^ capsules have a lower amount of released API at 120 min than hard gelatin samples.

## 4. Conclusions

This study aimed to develop a laboratory procedure suitable for preparing delayed-release capsules to encapsulate acid-sensitive substances. Conventional gelatin capsules or commercial DRcaps^TM^ based on HPMC and gellan gum without coating do not meet pharmacopeial limits for the delayed release. Therefore, both capsules were coated by immersion once in 10.0, 15.0, and 20.0% dispersions of Cellacefate, Eudragit^®^ S, and Acryl-EZE^®^. The plasticizer was not added to simplify the preparation process. In general, the coated gelatin capsules did not pass dissolution or disintegration tests regardless of the type of polymer coatings and the concentration of the dispersions. In most cases, Acryl-EZE^®^ coatings applied on DRcaps^TM^ capsules did not achieve acid resistance. DRcaps^TM^ capsules coated with 15.0% and 20.0% of Cellacefate released their content around the colon. However, all Eudragit^®^ S coated samples have proven suitable for achieving acid resistance. The results further show that the smoother surface of the DRcaps^TM^ capsule shell provides better adhesion to the polymeric coating than the hard gelatin capsule shell. This method is potentially feasible due to its simplicity to apply in the laboratory preparation of small samples containing acid-sensitive substances for experimental, clinical, or individual use. While the Eudragit^®^ S coating of DRcaps^TM^ capsules can be suitable for transport to the small intestine, Cellacefate coatings in higher concentrations applied to DRcaps^TM^ capsules can be intended to encapsulate substances for transport to the colon.

## Figures and Tables

**Figure 1 pharmaceutics-14-01577-f001:**
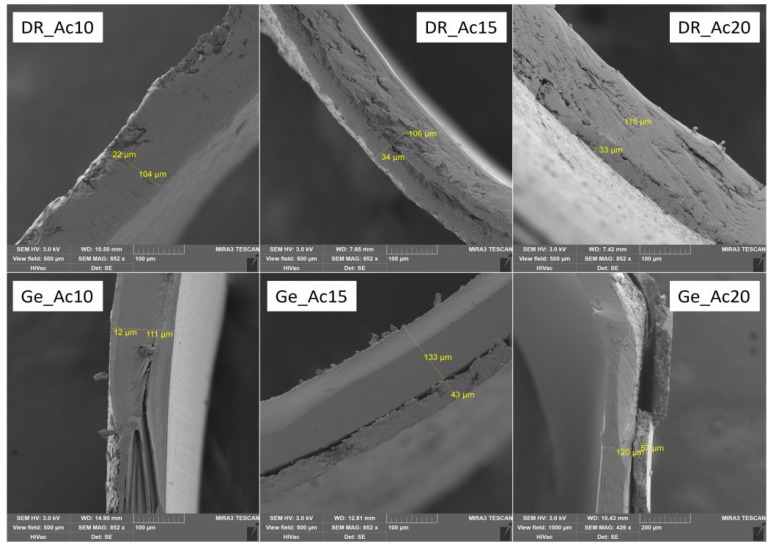
SEM images of selected samples with Acryl-EZE^®^ coating on both types of capsules.

**Figure 2 pharmaceutics-14-01577-f002:**
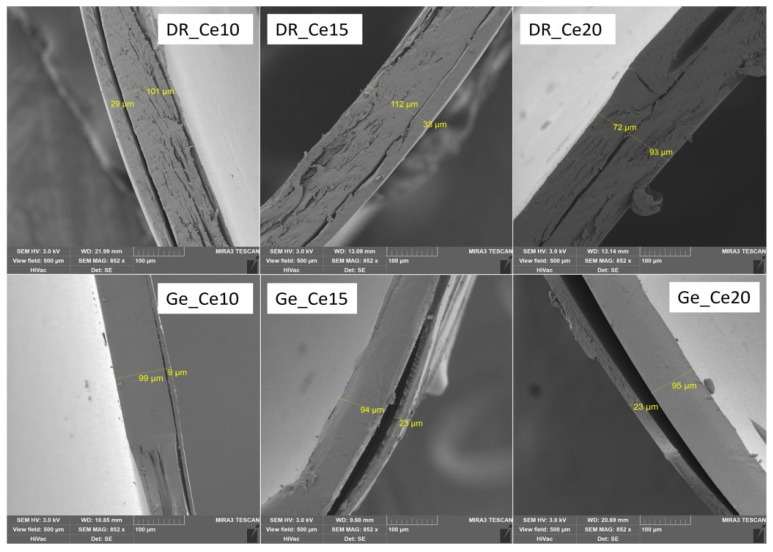
SEM images of selected samples with Cellacefate coating on both types of capsules.

**Figure 3 pharmaceutics-14-01577-f003:**
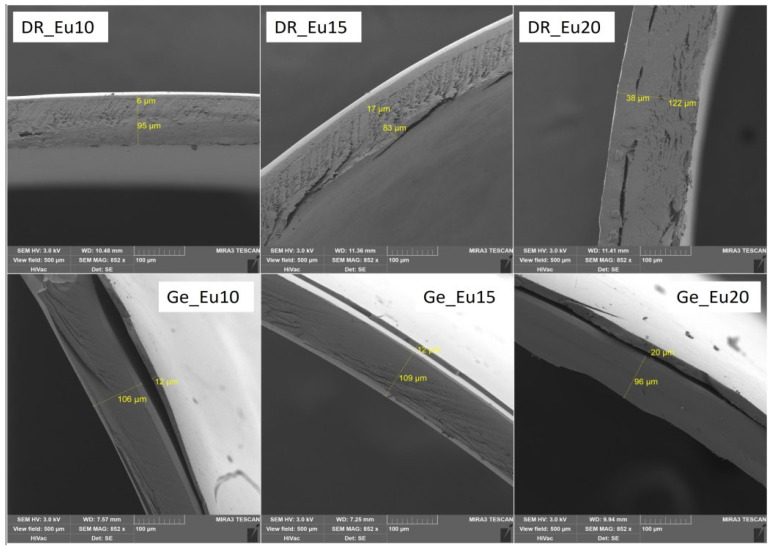
SEM images of selected samples with Eudragit^®^ S coating on both types of capsules.

**Figure 4 pharmaceutics-14-01577-f004:**
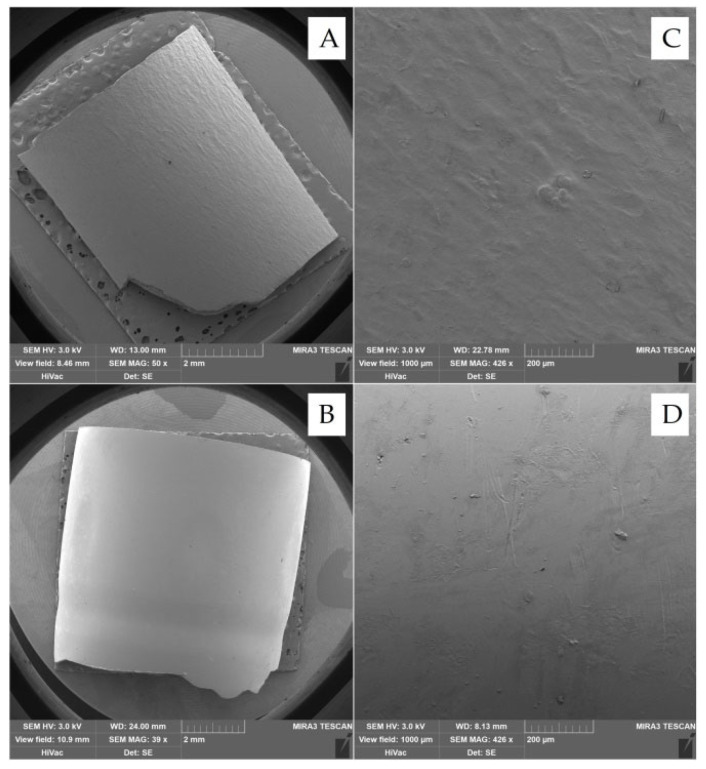
SEM images of the structure of the surface of capsules; (**A**) hard gelatin capsule; (**B**) DRcaps^TM^ capsule; (**C**) hard gelatin capsule (1000 µm view field); (**D**) DRcaps^TM^ capsule (1000 µm view field).

**Figure 5 pharmaceutics-14-01577-f005:**
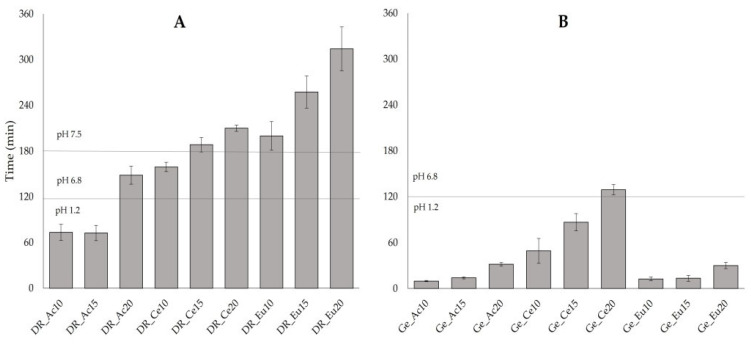
Time of disintegration under various conditions (pH 1.2; followed by the pH 6.8 and pH 7.5) of enteric-coated capsules: (**A**) DRcaps^TM^ capsules; (**B**) hard gelatin capsules. Mean values ± SD (*n* = 6).

**Figure 6 pharmaceutics-14-01577-f006:**
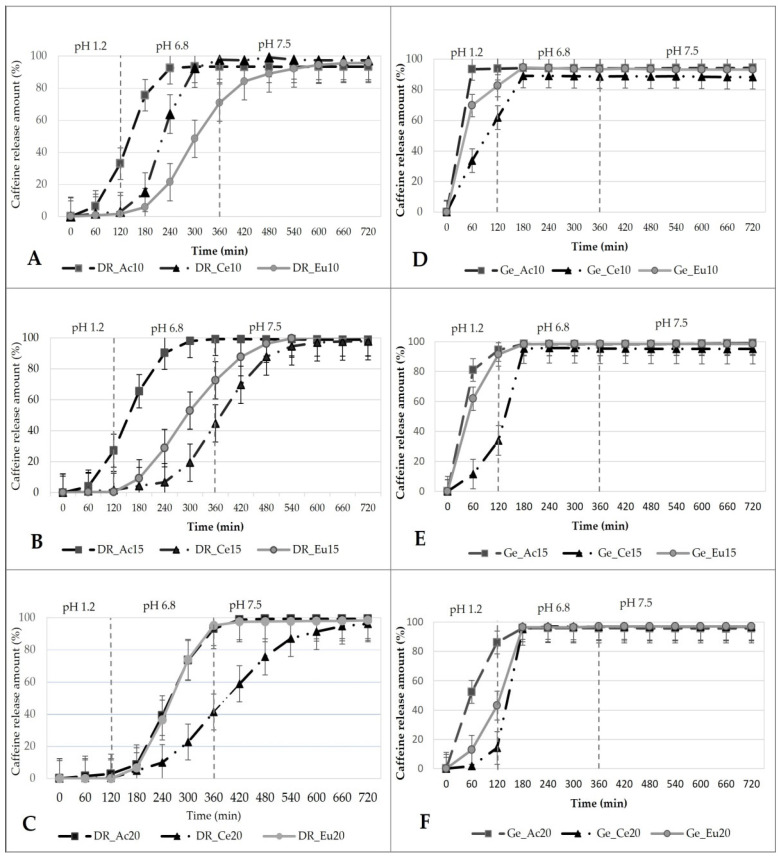
In vitro dissolution profiles of the enteric-coated capsules under three changing conditions (pH 1.2; pH 6.8; pH 7.5); (**A**) DR_samples with 10.0% polymer coatings, (**B**) DR_samples with 15.0% polymer coatings, (**C**) DR_samples with 20.0% polymer coatings, (**D**) Ge_samples with 10.0% polymer coatings, (**E**) Ge_samples with 15.0% polymer coatings, (**F**) Ge_samples with 20.0% polymer coatings.

**Figure 7 pharmaceutics-14-01577-f007:**
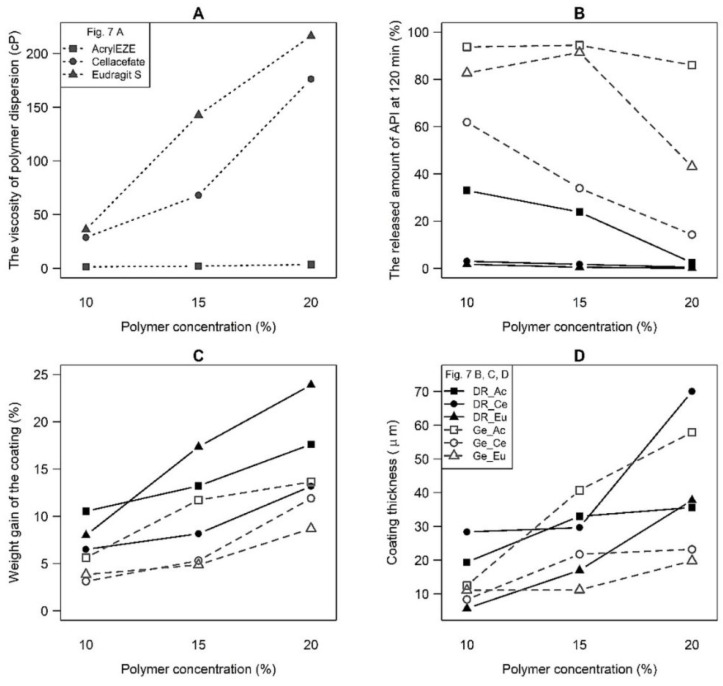
Interaction plot for (**A**) the viscosity of polymer dispersion, (**B**) the released amount of API at 120 min, (**C**) weight gain of the coating and (**D**) coating thickness; polymer types differentiated by symbols; DRcaps^TM^ samples (black, solid line) vs. hard gelatin capsules (grey, dashed line).

**Figure 8 pharmaceutics-14-01577-f008:**
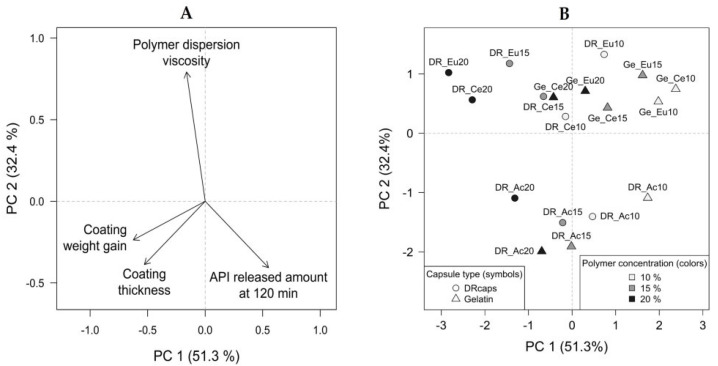
(**A**) PCA loadings plot; (**B**) PCA scores plot—capsule type differentiated by symbols, polymer concentration differentiated by colors, and polymer type differentiated by point labels.

**Table 1 pharmaceutics-14-01577-t001:** Characteristics of prepared hard capsule samples.

Sample **	Coating			The Practical Content of Caffeine per Capsule ± SD (mg)	Mean Weight of Uncoated Capsules ± SD (mg)
	Polymer	%	Solvent		
DR_Eu10	Eudragit^®^ S	10	Ethanol (96%)	101.50 ± 4.63	500.71 ± 10.96
DR_Eu15		15		98.40 ± 5.54	497.10 ± 9.38
DR_Eu20		20		103.40 ± 6.51	510.93 ± 3.16
DR_Ac10	Acryl-EZE^®^	10	Purified Water	99.20 ± 4.96	498.20 ± 10.05
DR_Ac15		15		102.10 ± 5.47	499.37 ± 10.60
DR_Ac20		20		95.60 ± 3.56	518.78 ± 8.58
DR_Ce10	Cellacefate	10	Acetone	97.80 ± 5.96	456.37 ± 4.49
DR_Ce15		15		103.40 ± 2.91	528.25 ± 7.97
DR_Ce20		20		99.30 ± 4.47	432.36 ± 6.45
Ge_Eu10	Eudragit^®^ S	10	Ethanol (96%)	102.80 ± 5.19	511.18 ± 11.03
Ge_Eu15		15		105.90 ± 5.82	509.95 ± 7.58
Ge_Eu20		20		101.10 ± 4.59	514.89 ± 7.55
Ge_Ac10	Acryl-EZE^®^	10	Purified Water	100.60 ± 5.43	501.83 ± 10.57
Ge_Ac15		15		103.30 ± 4.96	504.73 ± 8.52
Ge_Ac20		20		98.10 ± 4.87	520.10 ± 8.74
Ge_Ce10	Cellacefate	10	Acetone	103.90 ± 5.86	423.63 ± 9.05
Ge_Ce15		15		103.70 ± 5.59	529.19 ± 8.05
Ge_Ce20		20		105.50 ± 5.28	439.06 ± 8.29

Each sample consists of 20 capsules, with theoretical content of 100 mg of caffeine per capsule. ** Abbreviation of each sample was established by the type of capsule [Gelatin capsules (Ge); DRcapsTM (DR)]; the enteric polymer employed [Eudragit® S (Eu); Cellacefate (Ce); Acryl-EZE® (Ac)]; and the percentage of prepared dispersion.

**Table 2 pharmaceutics-14-01577-t002:** The viscosity of polymer dispersions with a constant temperature of 26.0 ± 0.5 °C.

Polymer/Solvent	Dispersion Concentration (%)	Viscosity ± SD (cP)	
Eudragit^®^ S/Ethanol 96% (*w*/*w*)	10	36.1 ± 1.25	
	15	142.8 ± 4.35	
	20	216.3 ± 6.60	
Acryl-EZE^®^/Purified Water	10	1.3 ± 0.12	
	15	2.0 ± 0.00	
	20	3.5 ± 0.00	
Cellacefate/Acetone	10	28.8 ± 2.16	
	15	67.9 ± 8.83	
	20	176.2 ± 21.23	

**Table 3 pharmaceutics-14-01577-t003:** Obtained weight of coatings samples dipped in different polymer dispersions. The thickness of polymer coating layers. Mean values ± SD (*n* = 20).

Sample	Mean Weight of the Coating ± SD (mg)	Mean Weight Gain of the Coating * ± SD (%)	The Average Thickness of Coating ± SD (μm)
DR_Eu10	7.99 ± 0.93	7.98 ± 1.21	5.70 ± 0.73
DR_Eu15	16.93 ± 2.90	17.36 ± 2.84	17.00 ± 2.15
DR_Eu20	23.78 ± 4.47	23.90 ± 4.81	37.75 ± 2.67
DR_Ac10	9.49 ± 2.44	10.53 ± 2.76	19.40 ± 3.79
DR_Ac15	13.17 ± 2.29	13.21 ± 2.58	29.0 ± 6.34
DR_Ac20	17.18 ± 1.80	17.60 ± 2.14	35.60 ± 7.97
DR_Ce10	6.28 ± 0.86	6.48 ± 0.96	28.35 ± 3.12
DR_Ce15	10.50 ± 0.99	8.15 ± 0.99	29.65 ± 4.96
DR_Ce20	18.49 ± 1.91	13.17 ± 2.10	70.00 ± 2.71
Ge_Eu10	4.28 ± 0.48	3.84 ± 0.50	11.10 ± 2.20
Ge_Eu15	5.37 ± 0.37	4.87 ± 0.42	11.20 ± 1.82
Ge_Eu20	9.85 ± 1.29	8.71 ± 1.39	19.80 ± 2.55
Ge_Ac10	5.47 ± 1.76	5.81 ± 1.90	12.50 ± 3.12
Ge_Ac15	11.18 ± 2.09	11.79 ± 2.31	40.70 ± 5.03
Ge_Ac20	15.09 ± 1.82	15.88 ± 1.94	57.90 ± 6.83
Ge_Ce10	3.39 ± 0.68	3.13 ± 0.73	8.35 ± 1.14
Ge_Ce15	7.40 ± 1.78	5.30 ± 1.91	21.75 ± 1.59
Ge_Ce20	18.44 ± 1.74	11.90 ± 1.89	23.20 ± 3.74

* Weight gain of coating relative to the total weight of the capsule.

**Table 4 pharmaceutics-14-01577-t004:** Comparison of dissolution profiles through similarity factor *f*_2_ and dissolution efficiency as ΔDE (%) expressed in 3 different levels, where two parameters are constant, and one is changeable: (**A**) Polymer concentration; (**B**) Polymer type; (**C**) Capsule type.

**(A) Polymer Concentration**					
Compared samples	*f*_2_/ΔDE (%)		Compared samples	*f*_2_/ΔDE (%)	
DR_Ce10 */DR_Ce15	15.28/−75.78		Ge_Ce10 */Ge_Ce15	35.32/−29.61	
DR_Ce10 */DR_Ce20	15.89/−73.47		Ge_Ce10 */Ge_Ce20	26.51/−52.17	
DR_Ce15 */DR_Ce20	79.44/9.53 **		Ge_Ce15 */Ge_Ce20	46.44/−32.05	
DR_Eu10 */DR_Eu15	66.12/11.35 **		Ge_Eu10 */Ge_Eu15	53.72/−0.64 **	
DR_Eu10 */DR_Eu20	37.87/48.4		Ge_Eu10 */Ge_Eu20	18.69/−45.69	
DR_Eu15 */DR_Eu20	42.12/33.27		Ge_Eu15 */Ge_Eu20	20.6/−45.34	
DR_Ac10 */DR_Ac15	46.25/−11.64		Ge_Ac10 */Ge_Ac15	28.04/−10.84	
DR_Ac10 */DR_Ac20	16.3/−47.28		Ge_Ac10 */Ge_Ac20	19.06/−26.76	
DR_Ac15 */DR_Ac20	22.49/−40.33		Ge_Ac15 */Ge_Ac20	36.31/−17.86	
**(B) Polymer Type**					
Compared samples	*f*_2_/ΔDE (%)		Compared samples	*f*_2_/ΔDE (%)	
DR_Eu10 */DR_Ce10	25.05/91.94		Ge_Eu10 */Ge_Ce10	28.78/−29.79	
DR_Ce10 */DR_Ac10	23.95/25.84		Ge_Ce10 */Ge_Ac10	14.23/74.88	
DR_Eu10 */DR_Ac10	13.34/141.54		Ge_Eu10 */Ge_Ac10	28.9/22.79	
DR_Eu15 */DR_Ce15	32.43/−58.25		Ge_Eu15 */Ge_Ce15	19.0/−50.26	
DR_Ce15 */DR_Ac15	12.3/359.04		Ge_Ce15 */Ge_Ac15	14.17/121.53	
DR_Eu15 */DR_Ac15	20.7/91.66		Ge_Eu15 */Ge_Ac15	45.77/10.19	
DR_Eu20 */DR_Ce20	22.62/−65.69		Ge_Eu20 */Ge_Ce20	40.66/−38.17	
DR_Ce20 */DR_Ac20	28.63/150.07		Ge_Ce20 */Ge_Ac20	17.39/167.8	
DR_Eu20 */DR_Ac20	52.19/−14.19 **		Ge_Eu20 */Ge_Ac20	24.91/65.6	
**(C) Capsule Type**					
Compared samples	*f*_2_/ ΔDE (%)	Compared samples	*f*_2_/ ΔDE (%)	Compared samples	*f*_2_/ ΔDE (%)
DR_Ce10 */Ge_Ce10	15.28/ 160.21	DR_Eu10 */Ge_Eu10	6.14/ 753.98	DR_Ac10 */Ge_Ac10	12.97/ 49.85
DR_Ce15 */Ge_Ce15	8.54/ 1546.18	DR_Eu15 */Ge_Eu15	6.22/ 557.71	DR_Ac15 */Ge_Ac15	13.63/ 83.04
DR_Ce20 */Ge_Ce20	10.42/ 1102.78	DR_Eu20 */Ge_Eu20	11.52/ 392.4	DR_Ac20 */Ge_Ac20	7.97/ 464.25

* Reference sample, ** Similar dissolution profiles.

**Table 5 pharmaceutics-14-01577-t005:** The ANOVA table for evaluating the entire data set is *p*-values (significant effects are indicated in bold).

Factor	Dispersion	Capsules		
	Viscosity	The Released Amount of API at 120 min	Weight Gain from the Coating	Coating Thickness
Polymer type (A)	**<0.001**	**<0.001**	**<0.001**	**<0.001**
Polymer concentration (B)	**<0.001**	**<0.001**	**<0.001**	**<0.001**
Capsule type (C)	NA	**<0.001**	**<0.001**	**<0.001**
A × B	**<0.001**	0.096	**<0.001**	**<0.001**
A × C	NA	**<0.001**	**<0.001**	**<0.001**
B × C	NA	**0.001**	**<0.001**	**<0.001**
A × B × C	NA	**<0.001**	**<0.001**	**<0.001**

NA—Not applicable.

## Data Availability

The datasets corresponding to the current study are available from the corresponding author upon request.

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
