# Peer review of "Development and Comparison of Various Coated Hard Capsules Suitable for Enteric Administration to Small Patient Cohorts"

_pharmaceutics, 2022, doi:10.3390/pharmaceutics14081577_

Round 1

Reviewer 1 Report

Please find the attached word document 

Author Response

Dear reviewer,

we would like to thank you for your factual comments, which we tried to incorporate or explain as best as possible. The attached document contains detailed responses to your comments.

Reviewer 2 Report

Ms.Nr: pharmaceutics-1814747

N Fülöpova et al: Development and comparison of various coated hard capsules suitable for enteric administration to small patient cohorts”

Authors present a systematic study aiming to investigate whether the simple immersion of capsules in different polymer dispersions is a successful method for achieving acid resistance and suitable disintegration and dissolution in the small intestine. 18 batches of enteric-coated capsules (DRcaps and hard gelatin) were prepared using Acryl-EZE, Cellacefate and Eudragit S polymers in10-15-20% concentrations and the relevant characteristic parameters were measured and compared with statistical methods. The topic is up-to-date and the experiments are well done. The conclusions are supported by the interpreted results.

I suggest the acceptance of the manuscript for publication after minor revision. Authors should address the critical comments below.

1. The commercial DRcaps are already modified and being enteric coated capsules but according to the literature data they do not always meet the Ph.Eur. requirements. Why authors did not used the original non-coated capsules as reference (blank) to demonstrate their drawbacks and the advantages of coating?

2. Why caffeine was selected as API? It is not an acid-sensitive molecule. Although it does not influence the conclusions about the enteric dissolution from capsules, a really instable drug would have been more convincing model compound.

3. On page 3, line 126: 470 mg lactose and 100 mg caffeine/ capsule is indicated, however, in Table 1 the mean weight of uncoated capsules varies from 423 to 528 mg. Contradiction must be corrected.

4. The15% and 20% Cellacefate coated formulations release the API at colon pH with more than 7h delay. Is there any drug where the colon can be the target of the absorption place? An example would be useful.

5. In table 3 the order of the 18 samples (Ac-Ce-Eu) is different than in table 1 and 2 (Eu-Ac-Ce). Is there any reason for this?

Author Response

(The authors gave the same response as above.)

Reviewer 3 Report

Page 1, Line 12: The problem stated is too generic and challenges established and continuously improving the technology of coating enteric capsules. And then there is a suddenly implied that the preparation of small batches is a problem. Authors need to restate the problem addressed.

Figure 6 needs to be explained in view of the gastric transit time. On one end, the delayed-release doesn't meet US pharmacopeial limits; while on the other end, the release rate is relatively slow. How can the benefit of this technology be achieved with these limitations? Please cite some possible examples in the discussion suitable for colon delivery.

It is difficult throughout the paper to track what Ge stands for. Authors are suggested to make a clear description of the sample key.

Author Response

(The authors gave the same response as above.)

Round 2

Reviewer 1 Report

The authors have responded to the comments/queries in an adequate manner and the same is incorporated in the revised manuscript.